# Clinical characteristics of treatment-resistant depression in adults in Hungary: Real-world evidence from a 7-year-long retrospective data analysis

Péter Döme[1,2☯], Péter Kunovszki[3☯]*, Péter Takács[3], László Fehér[4], Tamás Balázs[5], Károly Dede[5], Siobhán Mulhern-Haughey[6], Sébastien Barbreau[7], Zoltán Rihmer[1,2]

1 Department of Psychiatry and Psychotherapy, Semmelweis University, Budapest, Hungary, 2 Nyírő Gyula National Institute of Psychiatry and Addictions, Budapest, Hungary, 3 Janssen Global Commercial Strategy Organization, Budapest, Hungary, 4 Janssen-Cilag Ltd, Budapest, Hungary, 5 Healthware Ltd, Budapest, Hungary, 6 Janssen EMEA, Dublin, Ireland, 7 Janssen EMEA, Issy-les-Molineaux, France

☯ These authors contributed equally to this work.
* pkunovsz@its.jnj.com

**Data Availability Statement:** The dataset that was used is held by the National Health Insurance Fund (NHIF) of Hungary (http://www.neak.gov.hu, e-mail: neak@neak.gov.hu). Access to the individual-

## Abstract

### Purpose

Treatment-resistant depression (TRD) is associated with a poor quality of life and high economic burden. This observational retrospective epidemiological study aimed to estimate the proportion of patients with TRD within a cohort of patients with major depressive disorder (MDD) in Hungary and examine the mortality and comorbidities of patients with and without TRD.

### Patients and methods

This study included patients with MDD who experienced onset of a new depressive episode between 01 January 2009 and 31 August 2015, using data from a nationwide, longitudinal database.

### Results

Overall, 99,531 patients were included in the MDD cohort, of which 8,268 (8.3%) also met the criteria for TRD. The overall survival of non-TRD patients was longer than in TRD patients; the risk of mortality for TRD patients was significantly higher than of non-TRD patients (HR [CI] 1.381 [1.212–1.571]; p<0.001). Patients with TRD had a significantly higher probability of having "Neurotic, stress-related and somatoform disorders", autoimmune conditions, cardio- or cerebrovascular diseases, thyroid gland diseases and self-harming behaviour not resulting in death than non-TRD patients (for all comparisons, p values were less than 0.005).

level data is available after filing a formal data access request to adatkeres@neak.gov.hu. Requestors need to accept the terms and conditions of the data request and may need to pay the corresponding data access fee. The terms of the contract for data access does not allow the reporting of any data of a single individual or results which comes from aggregating the data of less than 10 individuals. Therefore, a de-identified dataset could not be provided. Taking these requirements into consideration, the results can be published. A supplementary dataset was created which contains the patient counts derived from the original data.

**Funding:** The study was funded by Janssen: Pharmaceutical Companies of Johnson & Johnson (www.janssen.com). PK, LF, SMH, SB are employees of Janssen, PT was an employee of Janssen during the time the study was carried out. Janssen provided funds for Healthware Ltd. for their participation in the study. TB, KD are employees of Healthware Ltd. Editorial assistance in the development of this manuscript was provided by Open Health Medical Communications (UK), with financial support from Janssen. The authors retained full editorial control over the content of the manuscript and the decision to publish it. The funder provided support in the form of salaries for authors PK, LF, SMH, SB, PT, provided funds for the data access, provided funds to Healthware Ltd., and provided funds for medical writing. The funder did not have any additional role in the study design, data collection and analysis, decision to publish, or preparation of the manuscript. The specific roles of these authors are articulated in the 'author contributions' section.

**Competing interests:** I have read the journal's policy and the authors of this manuscript have the following competing interests: PK, LF, SMH and SB are employees of Janssen. PT was formerly an employee of Janssen. TB, KD are employees of Healthware Ltd., which company received funding from Janssen for the participation in the study. PD received fees for consultancy from Janssen. ZR received fees for consultancy from Janssen, received speaker's honoraria from Janssen, Servier and Krka, and served as an advisory board member for Lundbeck, Janssen, Servier and Krka. This does not alter our adherence to PLOS ONE policies on sharing data and materials.

## Discussion

To our best knowledge, this is the first study to assess the frequency of TRD in Hungary. In a cohort of Hungarian MDD patients, we have found that the proportion of TRD (~8.3%) is comparable to those reported in previous studies with similar methodology from other countries. The majority of our other main findings (e.g. more frequent self-harming behaviour, increased risk of "Neurotic, stress-related and somatoform disorders" and higher overall mortality in TRD subjects) are also in line with previous results from other countries. Taking the substantial proportion of patients with TRD into consideration, a more comprehensive and targeted treatment strategy would be required for these individuals.

## Introduction

Major depressive disorder (MDD) is one of the most frequently occurring mental health disorders, with lifetime and 1-year prevalence of approximately 11–18% and 4–7%, respectively [1–6]. Consistent with results from other countries, a Hungarian study conducted in the mid-1990s found that the lifetime and 1-year prevalence of MDD was 15.1% and 7.1%, respectively [7]. MDD ranks highly in lists of global disability and burden of disease [2,4,5,8]. Findings from prospective studies suggest that the incidence of MDD and disability-adjusted life years associated with MDD are continuing to increase worldwide and is expected to become the leading cause by 2030 [3,6,9,10].

Pharmacotherapy-resistant depression (denoted as TRD in the following) is a term used to describe a subpopulation of MDD with a suboptimal response to an adequate dose and duration of antidepressants (ADs); however, there is not yet a standardised definition for TRD and existing definitions have key differences. For example, the number of unsuccessful AD trials in which patients experience a lack of response/remission, along with whether or not the subsequent AD should belong to a different AD class than the previous one, differs across TRD definitions [2,4,8,11–15], although most current definitions require the failure of at least two AD monotherapy trials [13,14]. The European Medicines Agency (EMA) defines TRD as a lack of clinically meaningful improvement despite the use of adequate doses of at least two AD agents belonging to the group(s) of commonly used first-line treatments prescribed for an adequate duration with adequate affirmation of treatment adherence [16].

Patients with TRD and non-TRD MDD differ from each other in several ways. Patients with TRD pose greater economic burden, than patients with non-TRD MDD, including higher outpatient medical costs, more frequent hospitalisations and, higher indirect costs [17–21]. Furthermore, health-related quality of life decreases with increasing levels of TRD [18]. It is worth mentioning, that MDD is a risk factor for the incidence of some somatic (e.g. cardiovascular) morbidities and positively associated with decreased life expectancy due to both natural (e.g. cardiovascular disorders) and unnatural (e.g. suicide) causes [22–27]. Furthermore, within the MDD group, patients with TRD also have higher all-cause mortality (particularly mortality from external causes, such as accidents and suicides) than patients with non-TRD MDD [13,28]. Of note, the background of suboptimal AD response cannot be confined to TRD, as in several cases the MDD itself is not refractory to AD treatment, rather other factors may contribute to the poor response to treatment. Such factors can include illness-related characteristics such as co-occurring somatic conditions (e.g. hypothyroidism, folate or vitamin $B_{12}$ deficiency, malignancies, Cushing's disease and Addison's disease) and a major depressive

episode in the context of undiagnosed bipolar I or II disorders, psychiatric comorbidities (e.g. anxiety and personality disorders) or patient-specific factors (e.g. non-compliance, insufficient plasma levels of ADs due to decreased absorption, drug interactions or pharmacogenetic peculiarities resulting in enhanced metabolism of the given AD) [2,4,8,19,29–37]. However, it appears that the adequate treatment of at least some somatic comorbidities, such as hypothyroidism, results in the dissipation of the difference in AD responses between patients with and without the given somatic comorbidity [38].

Some features of an MDD episode may also be associated with treatment resistance; for example, greater symptom severity, psychotic symptoms, suicidality and longer duration of the current episode are predictors of treatment resistance [8,11,37]. In the STAR*D trial, remission rates were highest in the first two AD trials and dropped significantly in subsequent AD trials, and even for the small number of patients with TRD that do achieve remission, 60% relapsed within 6 months [39,40]. Another study found that higher numbers of previously administered ADs (for the current and previous episodes) were positively correlated with the risk of treatment resistance [8]. However, the vast majority of studies show not only overt, but also 'subthreshold', bipolarity (marked by, for example, agitated/mixed depressive episodes, early onset, depressive episodes with psychotic/atypical/anxious features, family history of bipolar disorder, highly recurrent [i.e. more than three] depressive episodes), may be associated with poorer response to AD treatment [8,11,29,30,32,41–48].

Despite the use of different protocols and TRD definitions, the proportion of patients with TRD is typically higher in studies based on clinical datasets (approximately 35% of patients in the STAR*D and European Group for the Study of Resistant Depression trials failed to achieve a response after two consecutive AD trials [add-on treatment was also given in some cases in both trials]) than in studies based on administrative (prescription database or registry) data (approximately 4–20%) [11,13,20,37,39,49–54]. Due to differences in healthcare systems, available data structure and applied TRD definitions, highly variable approaches have been used in studies based on administrative data from different clinical settings and countries. The majority of these studies defined a case as treatment resistant if a minimum two AD treatment regimens (typically with an AD, but in some studies the use of add-on medications was permitted) had failed. Failures were mainly defined as a switch from the current treatment regimen and/or beginning an add-on medication (e.g. another AD or an antipsychotic), or discontinuing the regimen [13,49–53].

As confirmed by a review of relevant literature, no studies to date have investigated the prevalence of TRD in Hungary; therefore, the current study used data from the nationwide, longitudinal financial claims database of the Hungarian National Health Insurance Fund (NHIF) to analyse the characteristics of a TRD cohort within a newly identified MDD population in Hungary.

The primary objective of this study was to estimate the proportion of patients with TRD in the MDD cohort investigated. Secondary objectives were to identify the frequency of conditions thought to be associated with TRD, such as other psychiatric and somatic comorbidities, and self-harming behaviour not resulting in death. We also sought to detect differences in the demographic profile, treatment, and mortality between TRD and non-TRD populations in Hungary.

## Materials and methods

### Data source

This observational, retrospective, epidemiological study used data from the nationwide, longitudinal database of the NHIF. This includes detailed healthcare service data for the entire

population of Hungary (approximately 10 million people). All recorded healthcare events are linked to individual patients by a unique patient identifier (social security number), thereby allowing longitudinal patient pathway analysis. The database includes demographic data, such as date of birth, gender, and geographical region, for all patients. It also has financial claims information for all inpatient hospital stays and outpatient visits reimbursed by the insurance fund from 2009 to the present, along with drug dispensation information. Medication prescription data include diagnosis based on ICD-10 codes, date of dispensation, brand/generic names, dosages and formulations, intended route of administration, and number of tablets and injections. Patient death information is also included along with the date of death of deceased patients. Reporting of deaths is rigorous, practically no missing data can be expected in deaths.

By law, the NHIF has a right to handle patients' data (Act no. 80/1997 on mandatory health insurance coverage) and share the data on a claim basis (Act no. 63/2012 on the re-use of public data). In the current study, only the NHIF had direct access to patient-level data. Other members of the research group accessed the data indirectly through the NHIF in accordance with their data privacy regulations, therefore patient consent for this analysis was not required.

## Patient selection

**Timeframe.** Patient data were analysed for the period between 01 January 2009 and 31 August 2015. Data were also available between 01 January 2007 to 31 December 2008; however, due to changes in data access regulations, these data could not be analysed in detail, but only used to identify patients with MDD/TRD from 2009 onwards.

This study included patients with MDD who experienced onset of a new depressive episode between 01 January 2009 and 31 August 2015. However, this should not be considered as an incident cohort; due to the recurrent nature of depression, it is possible that some patients may have experienced a depressive episode prior to 2007.

**Definition of MDD patient cohort.** The MDD patient population was selected in a way that could ascertain TRD with as little ambiguity as possible. For this purpose, patients with certain conditions that could easily interfere with the process were excluded from the study. Therefore, while not all patients with MDD were captured in the study, the TRD status of those in the study could be obtained with a higher degree of certainty.

AD prescriptions were identified through Anatomical Therapeutic Chemical (ATC) codes and then used to find patients with a diagnosis of MDD based on the presence of ICD-10 codes F32 and F33. Patients were included in the MDD population in this study if they had at least two AD prescriptions within 6 months at any time during the study, with the first AD prescription taken as the MDD index date for the patient (Fig 1). Patients were only selected if they had no AD prescriptions or MDD diagnoses in the period between 01 January 2007 and 31 December 2008, thus all patients had at least a 2-year period with no AD prescriptions prior to the MDD index date.

Patients were excluded based on diagnoses (S2 Table in S2 File) and medication prescriptions (S3 Table in S2 File). A patient was excluded if they had at least two instances of the same ICD-10 diagnosis code from the excluded diagnoses list within 180 days of each other in the period starting 180 days before the MDD index date and ending at the end of follow-up. A patient was also excluded if they had any prescriptions for medications from the excluded medication list within 180 days preceding the MDD index date.

The rationale for the exclusion of patients with some somatic and substance use disorders that can potentially lead to depression was two-fold: firstly, the Diagnostic and Statistical Manual of Mental Disorders requires that, in a patient with MDD, the depressive episode would

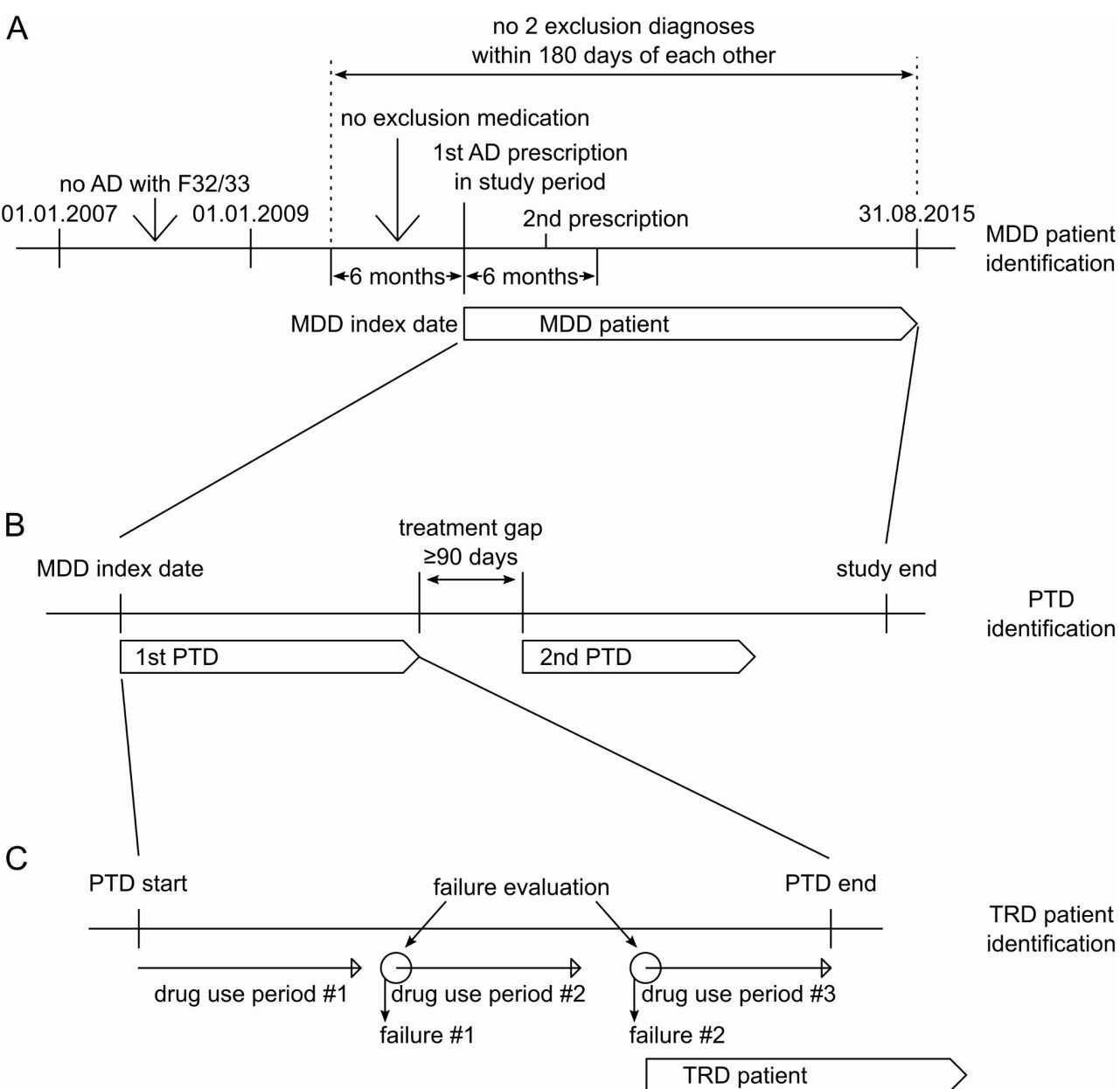

**Fig 1. Schematic of patient inclusion and determination of TRD.** The treatment pattern of patients was considered on three different levels. Individual prescriptions were compiled into drug use periods, which were then compiled into PTD episodes. Then, the whole follow-up period for patients was broken down stepwise to determine TRD. Step A–MDD patient identification: Patients are identified based on the inclusion and exclusion criteria. All patients with AD prescriptions in 2007 and 2008 are excluded. Step B–PTD identification: The PTD episodes needed to be identified as treatment failures were only considered within a single PTD episode. Step C–TRD determination: The failure of a treatment (corresponding to a drug use period) was evaluated at the start of the subsequent treatment (regardless of the previous treatment having ended or not). After the second failure was identified (within a PTD episode), the patient was identified to have TRD (up to the end of follow-up). AD, antidepressant; MDD, major depressive disorder; PTD, pharmaceutically treated depression; TRD, treatment-resistant depression.

not be attributable to the physiological effects of a substance or another medical condition [55]; secondly, as has already been mentioned in the 'Introduction' section, co-occurring somatic conditions may worsen treatment responses to ADs, and treatment of the given somatic disorder may improve the efficacy of ADs. Patients with these somatic conditions

were excluded in an attempt to create a cohort in which the response rate to ADs is independent of the presence or absence of adequate treatment for the given somatic comorbidity. Some non-somatic (i.e. psychiatric) conditions were also causes of exclusion (e.g. manic episode [to exclude bipolar depressive patients] or schizoaffective disorder). Using these exclusion criteria, we have excluded patients whose depressive symptoms/episodes were present in the context of a psychiatric disorder other than MDD. The excluded medication list comprises drugs (mainly psychotropic agents) for which use is indicative of the presence of a psychiatric/neurological disorder that is either comorbid with MDD or can potentially lead to symptoms of depression (e.g. the administration of mood stabilisers is indicative of the presence of bipolar disorder).

## Study design

TRD status was assigned based on treatment patterns. Drug use periods (drug purchases with the same ATC code) were created for all ADs and potential add-on medications using an algorithm similar to the PRE2DUP method, with a 30-day grace period of non-treatment allowed within the drug use period [56]. Drug use periods were then compiled into pharmaceutically treated depression (PTD) episodes. These episodes started when a patient began treatment with an AD medication (the first therapy could not be an add-on medication) after at least 90 days of no treatment and lasted until the patient stopped all treatment, followed by at least 90 days of no treatment.

Treatment failures were used to determine TRD status. A patient was considered to have TRD if two different treatments had failed during a PTD episode (Fig 1). The date of the second treatment failure was taken as the TRD index date for the patient.

In addition, an algorithm similar to the one found in Li et al. [28] was used to capture treatment failures in cases of consecutive initiations of different treatments within the same PTD episode. Fig 2 shows the different scenarios in which a treatment can be evaluated and the result (treatment failure or not).

## Study measures

Demographic data for age, sex and follow-up time were collected for patients with and without TRD. Data collected included the proportion of patients using drugs from different drug classes (ADs [tricyclic antidepressants, selective serotonin reuptake inhibitor (SSRI), serotonin–norepinephrine reuptake inhibitor, monoamine oxidase inhibitors, other; for a list of all medications considered see S4 Table in S2 File] and add-on drugs [atypical antipsychotics, antiepileptics, lithium, buspirone and thyroid hormone; for a list of all medications considered see S5 Table in S2 File]) and the proportion of patients using the 10 most commonly prescribed drugs.

The number of new patients per year in the overall MDD cohort (based on the MDD index date) and in the TRD sub-population (based on the TRD index date) were reported. All patients were followed-up until 31 August 2015 or their death if this occurred sooner than the follow-up end date.

Mortality was assessed using survival analysis with a matched model, in which patients with TRD were matched to those without TRD using incidence density sampling. Using this technique, TRD patients are matched to other MDD patients who are not TRD at the same time (measured from their MDD index date), this date is denoted as "matched date" in the following. If TRD patients had only been matched to non-TRD patients a selection bias would have arisen due to excluding patients based on future events (the patients who are not yet TRD but become one at a later time). Three patients without TRD were matched to each patient with

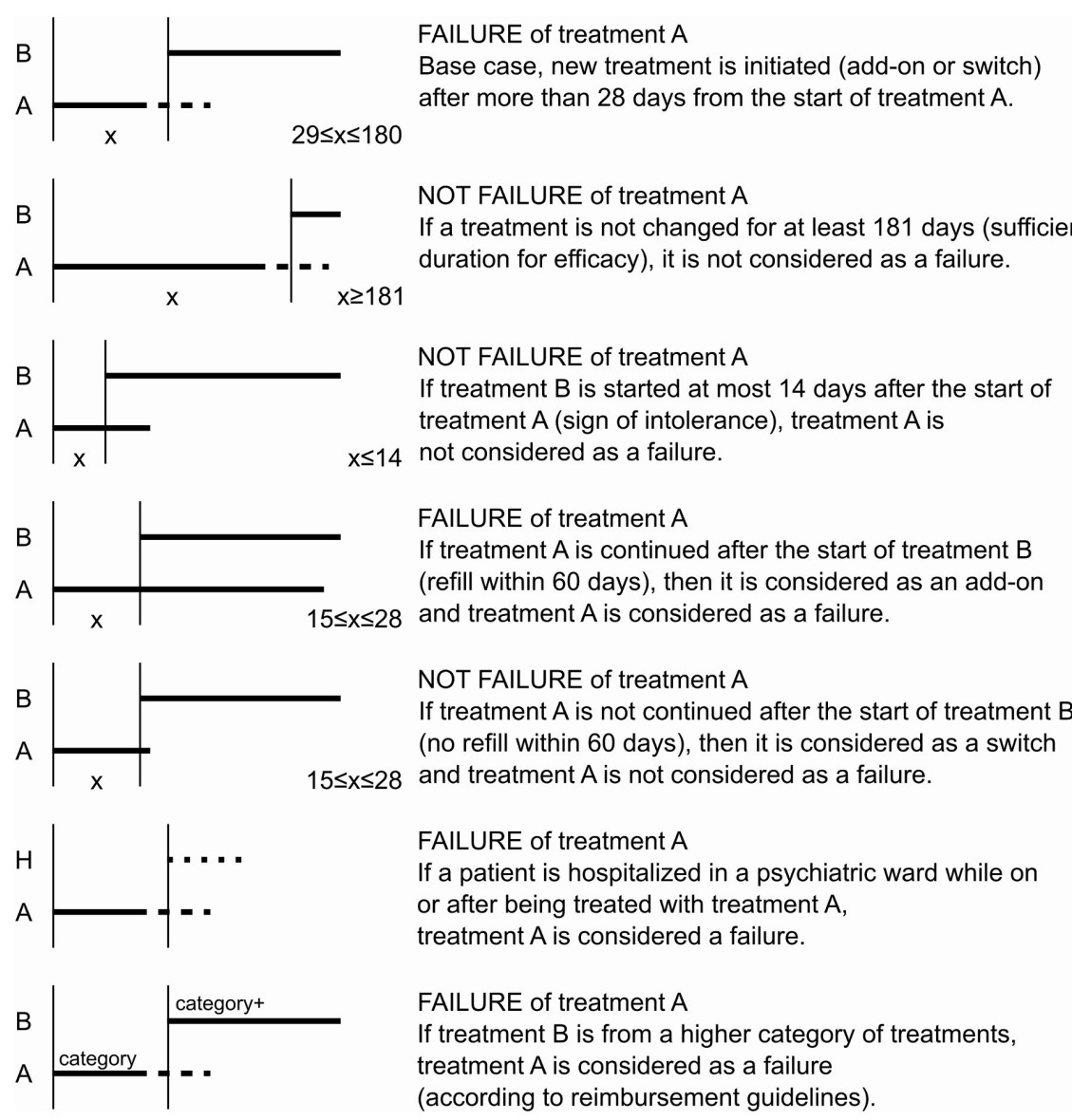

**Fig 2. Algorithm for treatment failure evaluation.** The figure illustrates all the possible scenarios for evaluating of failure of a treatment (A). The evaluation occurs at the start of the next event [initiation of the next treatment (B) or hospitalisation in a psychiatric ward (H)]. The dashed line at the end of treatment A illustrates that the evaluation result was independent of whether treatment A was being received at this time point. X represents the number of days elapsed. The category of treatments (on a scale of 1–3) was determined based on the Hungarian reimbursement guidelines (see S7 Appendix in S3 File).

TRD based on age and gender. The start date for the survival analysis was the TRD index date for TRD patients. For the matched patients the "matched date" defined above was used as the start date. Only patient death was considered as an event, the date of which was obtained from the database as it was available directly. If a matched patient became treatment resistant during follow-up, that patient was censored at the time treatment resistance occurred. Otherwise patients were only censored at the end of follow-up (on 31 August 2015). As patients were only matched on age and gender, the estimate of the effect of TRD on mortality was confounded by other effects, such as comorbidity profile, smoking and other lifestyle factors, among others.

A prespecified list of comorbidities was analysed in a fixed 2-year time period, which started on the TRD index date for patients with TRD and on the MDD index date for patients without TRD. Analysed comorbidities included "Neurotic, stress-related and somatoform disorders", thyroid disorders, malignancies, autoimmune conditions, cardio- or cerebrovascular disease, and self-harming behaviour not resulting in death (a full list of ICD-10 codes for the comorbidities analysed can be found in S6 Table in S2 File). Patients with a corresponding index date later than 31 August 2013 were excluded from the analysis. Patients who died during the 2-year time period (but otherwise could have been observed for the whole 2 years) were not excluded. Presence of a comorbidity was defined as the appearance of the corresponding diagnosis code at least twice within inpatient records, outpatient records or on prescriptions within 90 days, except in the case of self-harming behaviour where a single record was sufficient. To construct this category, all the ICD-10 codes that reflect explicit self-harming intent (X60-84) were included, as were some forms of self-harming with undetermined intent and various forms of poisoning (see S6 Table in S2 File).

## Statistical analyses

Descriptive statistics were used to present patient demographics, epidemiology, and medication use. Male and female proportions were compared using a chi-square test of equal probabilities.

A Kaplan–Meier estimator was used to estimate overall survival. The Cox proportional hazards model was used to compare the survival of patients with and without TRD; age and gender were also included in the model for adjustment purposes.

The risk of comorbidities being present was analysed using a logistic regression. Similarly to the Cox model, the effect of interest was TRD status (TRD/non-TRD), while age and gender were included as adjustment variables. Odds ratios (OR) for TRD vs. non-TRD were reported for all models together with a p-value for a z-test for no effect (OR = 1).

Age was used as continuous variable in the Cox and logistic models with the assumption of a linear effect.

## Ethical approval

This study has been approved by Medical Research Council–Research and Ethics Committee (TUKEB), Hungary (Appr. no: 24431-2/2016/EKU (original), 44246-2/2018/EKU (extension)).

All data used in the study was held by NHIF, the researchers had access only to anonymized data. Data protection guidelines did not permit the reporting of patient level data even in anonymized form, only aggregate results could be reported.

The database contains information on the full population of Hungary (approximately 10 million people). The study population consisted of 99,531 patients after applying all inclusion and exclusion criteria.

## Results

A total of 99,531 patients with MDD were identified during the study period, of which 8,268 (~8.3%) met the criteria for TRD. Patient demographics are presented in Table 1.

Whilst women represented 66.6% of the total study population, a significantly larger proportion of the TRD population were women (71.0%) compared with the non-TRD population (66.2%; p<0.0001). The mean age of the non-TRD and TRD subgroups was similar (49.6 years and 48.4 years, respectively).

**Table 1. Patient demographics.**

| Total patients with MDD, n | 99,531 | |
|---|---|---|
| | **Non-TRD** | **TRD** |
| Total patients with MDD, n | 91,263 | 8,268 |
| Women[a], n (%) | 60,427 (66.2) | 5,872 (71.0) |
| Age (mean), years | 49.6 | 48.4 |
| Age groups, n (%) | | |
| 18–19 | 1,144 (1.3) | 61 (0.7) |
| 20–29 | 9,706 (10.6) | 700 (8.5) |
| 30–39 | 17,725 (19.4) | 1,624 (19.6) |
| 40–49 | 17,386 (19.1) | 1,913 (23.1) |
| 50–59 | 20,793 (22.8) | 2,296 (27.8) |
| 60–69 | 11,942 (13.1) | 1,007 (12.2) |
| ≥70 | 12,567 (13.8) | 667 (8.1) |
| Follow-up time (days)[b] | | |
| Mean (SD) | 1,266 (711) | 1,515 (628) |
| Median | 1,300 | 1,621 |

[a] Difference is significant at $p < 0.0001$.

[b] The period between the first relevant ICD-10 registration and death date/study end date.

## Number of new MDD and TRD cases

The total number of new MDD and TRD cases occurring between 1 January 2009 and 31 August 2015 is presented in Fig 3.

The number of new patients identified decreased year on year from 19,049 in 2009 to 12,963 in 2014. The decrease in new MDD cases over the study period was an artefact caused by the study design, which can be attributed to the increasing duration of the AD-free period prior to the MDD index date (Fig 1). This phenomenon is explained in the study limitations

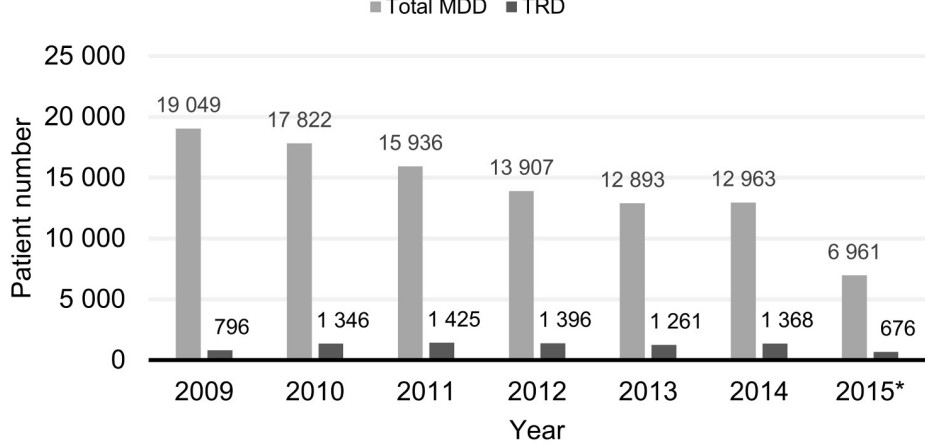

**Fig 3. Number of new MDD and TRD cases by year.** New MDD patients were identified at the patient's first AD prescription (MDD index date). New patients with TRD were identified at the date of the second treatment failure within the same PTD (TRD index date). The same patient could have the MDD index date and TRD index date in different years. *Data for 2015 only available up to 31 August 2015. AD, antidepressant; PTD, pharmaceutically treated depression; MDD, major depressive disorder; TRD, treatment resistant depression.

section. A relatively low number of newly identified MDD cases (6,961) were identified in 2015 as data were available only up to 31 August 2015.

The number of newly identified TRD cases between 2010 and 2014 was relatively consistent and ranged from 1,261 to 1,425 per year. A total of 796 new cases of TRD were identified in 2009; however, this was again a low number due to the relatively small patient pool available for 2009 compared with other years (Fig 1). Only 676 new TRD cases were identified in 2015 due to the data cut-off date. We found that the TRD subgroup comprised approximately 8.3% of the total MDD cohort.

For the purposes of comparison, it should be noted that during the study period, the adult population of Hungary remained relatively stable at approximately 8.1 million people [57].

## Mortality

Each patient with TRD (n = 8,268) was successfully matched to three patients without TRD (n = 17,516) based on gender and age.

The estimated Kaplan–Meier survival curves with 95% confidence bands are presented in Fig 4; this shows that overall survival of patients without TRD was longer than patients with TRD. Furthermore, the risk of mortality for patients with TRD, identified using the age and gender-adjusted Cox regression model, was significantly increased overall compared with patients without TRD: hazard ratio (HR [CI]) 1.381 (1.212–1.571); p<0.001.

## Comorbidities

A total of 68,167 patients without TRD and 5,822 patients with TRD were included in the comorbidities analysis.

Age- and gender-adjusted logistic regression analysis revealed that patients with TRD had a significantly higher probability of having "Neurotic, stress-related and somatoform disorders", autoimmune conditions, cardio- or cerebrovascular diseases, thyroid gland diseases and self-harming behaviour not resulting in death compared with patients without TRD (Table 2). There was no significant difference in the probability of having malignancies between the two groups.

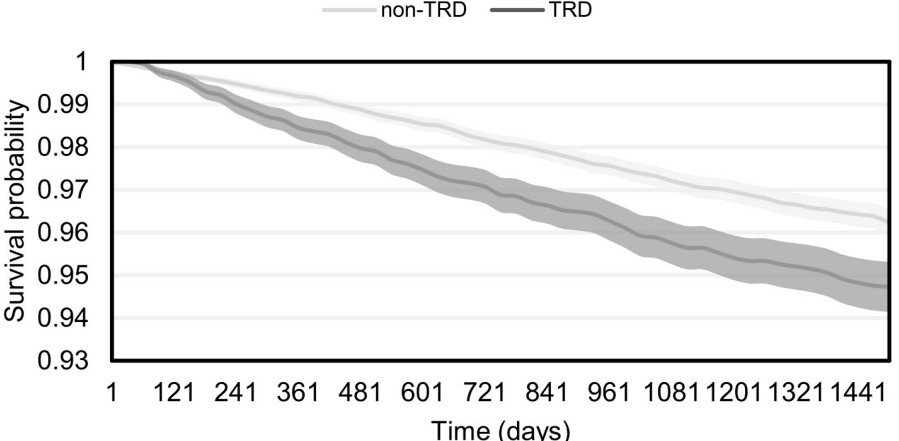

**Fig 4. Overall survival of TRD and non-TRD patients.** Start: TRD index date for patients with TRD, "matched date" for non-TRD patients; event: death; censoring: at the study end (31 August 2015) or when matched non-TRD patients develop TRD. TRD, treatment-resistant depression.

**Table 2. Comorbidities.**

| Comorbidity, n (%) | non-TRD n = 68,167 | TRD n = 5,822 | OR | 95%CI | p |
|---|---|---|---|---|---|
| "Neurotic, stress-related and somatoform disorders" | 19,966 (29.3%) | 2,512 (43.1%) | 1.81 | 1.71–1.91 | <0.0001 |
| Malignant disease | 2,148 (3.2%) | 153 (2.6%) | 0.93 | 0.79–1.10 | 0.4205 |
| Autoimmune disease | 2,605 (3.8%) | 260 (4.5%) | 1.22 | 1.07–1.39 | 0.0025 |
| Cardio- or cerebrovascular disease* | 9,687 (14.2%) | 783 (13.4%) | 1.17 | 1.08–1.28 | 0.0002 |
| Self-harming behaviour not resulting in death | 1,732 (2.5%) | 233 (4.0%) | 1.58 | 1.38–1.82 | <0.0001 |
| Thyroid gland disease | 751 (1.1%) | 92 (1.6%) | 1.42 | 1.14–1.77 | 0.0017 |

This analysis is restricted to patients with an MDD index date (for non-TRD) or TRD index date of 31 August 2013 or earlier to allow for a 2-year follow-up period. Number and percentage of patients with comorbidities in the TRD and non-TRD group; OR obtained from age and gender-adjusted logistic regression analysis; 95% confidence interval for OR; p-value for the test OR = 1.

*Note that whilst a higher frequency of cardio- and cerebrovascular disorders was observed in the non-TRD group, the risk of these comorbidities was found to be higher in the TRD group using the age- and gender-adjusted model. CI, confidence interval; MDD, major depressive disorder; OR, odds ratio; TRD, treatment-resistant depression.

It should be noted that whilst a higher frequency of cardio- and cerebrovascular disorders was observed in the non-TRD group, the risk of these comorbidities was higher in the TRD group than in the non-TRD group using the age- and gender-adjusted model.

## Treatment patterns

To assess the performance of the TRD decision algorithm, data regarding the number of different treatments (ADs and add-on therapies) prescribed to patients for the whole study period were collected. Based on the study exclusion/inclusion criteria for non-TRD and TRD, the majority of patients without TRD were prescribed either one (41%) or two (31%) treatments during the study period, which demonstrated that the algorithm worked as expected. A total of 28% of patients without TRD received more than two treatments. It should be noted that the number of different treatment used were counted for the whole study period and not within a single PTD episode; therefore, if patients without TRD experienced several major depressive episodes during the study, they could be prescribed different treatments for each PTD episode, provided that no more than one treatment within each PTD episode was considered to have failed, the patient would still be classed as non-TRD. In the TRD population, most patients received three or more different treatments (86.9%; Fig 5).

The most commonly prescribed drug class in the study overall was SSRIs (Fig 6). For patients without TRD, citalopram (29%), escitalopram (25%) and paroxetine (22%) were the most commonly prescribed AD treatments; and for patients with TRD, mirtazapine (51%), escitalopram (44%) and venlafaxine (37%) were the most commonly prescribed (Fig 7). For the list of medications belonging to each category see S4 and S5 Tables in S2 File.

## Discussion

To date, there has been a lack of consensus on the definition of TRD, in particular, regarding the number of unsuccessful AD trials required before MDD is classed as TRD. Different studies have based their definition of TRD on varying numbers of unsuccessful AD trials, although it should be noted that these studies differed not only from their definition of TRD, but also from the specific study settings used [2,4,8,11–15].

The primary objective of this study was to estimate the proportion of patients with *pharmacotherapy-resistant depression* in the cohort investigated using data from a nationwide, longitudinal database. Secondary objectives were to compare the TRD and non-TRD MDD

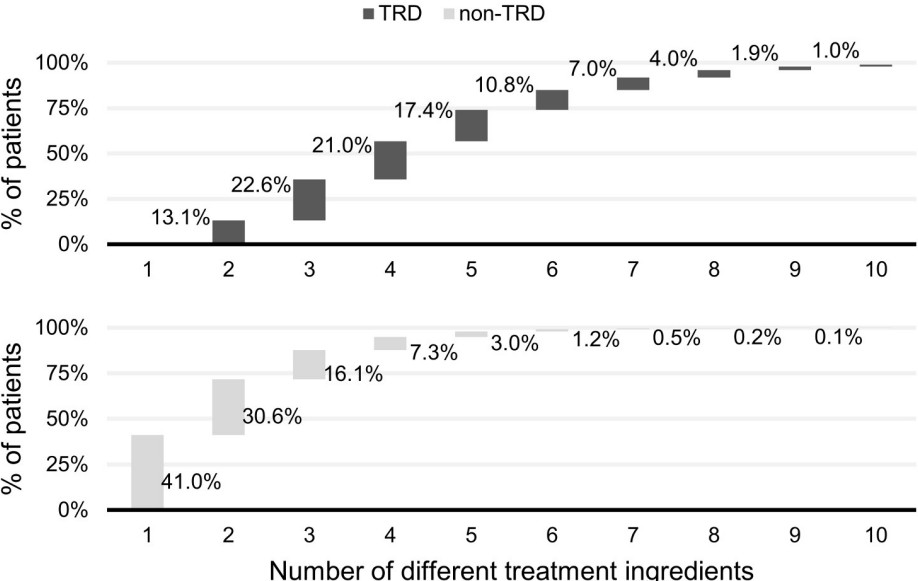

**Fig 5. Distribution of patients by the number of different treatment ingredients used throughout the whole study period.** This analysis includes antidepressant and add-on therapies. Only the number of different treatment ingredients was counted, so a patient using the same treatment multiple times would only count as one treatment. The numbers are counted within the whole study period (and not within a single PTD episode). PTD, pharmaceutically treated depression.

populations in Hungary with regards to the following: demographic profile; treatment patterns; mortality rate; frequency of patients with specific psychiatric and somatic comorbidities, and self-harming behaviour not resulting in death.

We found that the proportion of patients with TRD, as defined in this study, within our MDD cohort was approximately 8.3%, which falls within the range of estimates (4–20%)

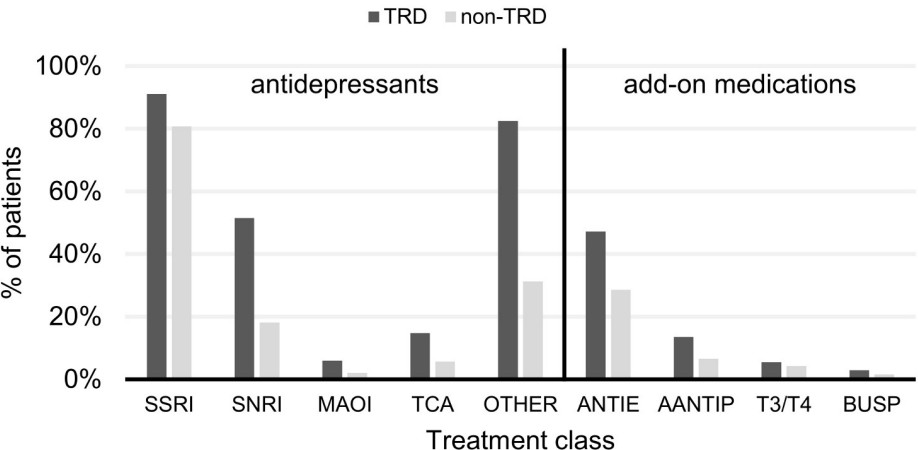

**Fig 6. Use of specific drug classes during the whole study period.** Both antidepressant treatments and add-on therapies are included. A patient was considered to use a certain class of treatment if they had at least one prescription filled from the given class of treatment. Lithium is not included in the figure due to very low patient counts. 'Other' includes other antidepressant treatments not included in the classes stated. AANTIP, atypical antipsychotics; ANTIE, antiepileptics; BUSP, buspirone; MAOI, monoamine oxidase inhibitors; SNRI, serotonin-norepinephrine reuptake inhibitors; SSRI, selective serotonin reuptake inhibitors; T3/T4, thyroid therapy; TCA, tricyclic antidepressants.

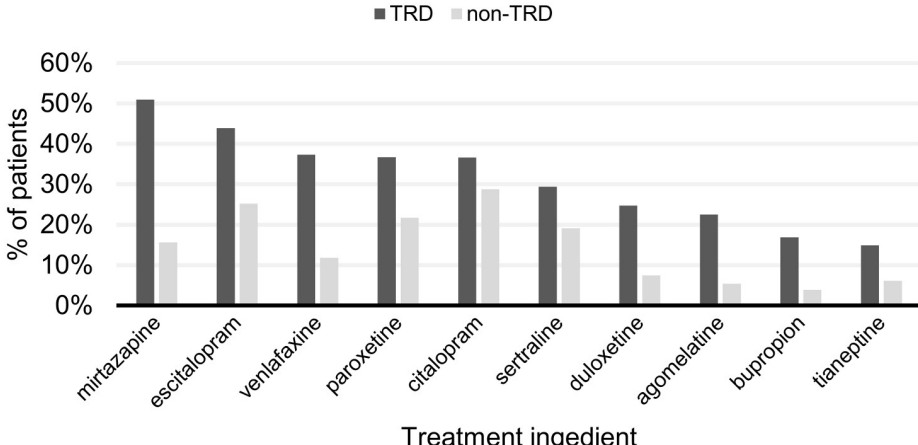

**Fig 7. Top 10 most frequently used antidepressant treatments during the whole study period for non-TRD and TRD patients.** Treatments are presented in descending order by the frequency of patients with TRD using them. For patients with TRD, the antidepressants used also include those used prior to identification of TRD. TRD, treatment-resistant depression.

reported in previous studies from other countries using similar methodology, i.e. observational studies using prescription databases or registries [11,13,49–52]. It should be noted that studies based on clinical datasets tend to report higher rates of TRD due to the use of specific rating scales to measure clinical response parameters [37].

The non-TRD and TRD populations differed in terms of gender, but not age; with the TRD population comprising a higher proportion of women compared with the non-TRD population. Whilst depression is known to be more common in women than men [5], previous studies have been unable to demonstrate any clear relationship between gender and treatment resistance in depression [35,37,49–51,58].

This study investigated the association between MDD, TRD and a prespecified list of comorbidities, namely, "Neurotic, stress-related and somatoform disorders", thyroid disorders, malignancies, autoimmune conditions, cardio- or cerebrovascular disease, and self-harming behaviour not resulting in death. Some of these comorbidities were more strongly associated with TRD than non-TRD MDD. In terms of psychiatric comorbidities, the TRD population was observed to have an increased risk of comorbid "Neurotic, stress-related and somatoform disorders" (ICD-10 F40–F48; see S6 Table in S2 File) compared with the non-TRD population. This finding is in line with the results of several previous studies [2,8,20,35,39,49,50,58]; however, Perugi et al. reported an almost identical prevalence of anxiety disorders in patients with MDD with and without TRD [29]. Conversely, Gronemann et al. reported that patients with depression with comorbid anxiety had a lower risk of TRD compared with patients without comorbid anxiety, after adjustment for other measures of depression severity [54]. The presence of mixed symptoms of anxiety and depression could indicate bipolar disorder. As depressive episodes in the context of bipolar disorder are less responsive to AD treatment compared with those in MDD, the inclusion of patients with unidentified bipolar (spectrum) disorder could potentially distort the frequency of TRD [19,59–61]. With regard to somatic comorbidities, previous studies have reported conflicting results [8,18,34,35,37,39,49,50,58,62,63]. Within this study, we detected a significantly higher risk of autoimmune diseases, thyroid gland diseases, and cardio- and cerebrovascular disorders in patients with TRD compared with the non-TRD group.

The importance of using age- and gender-adjusted models when assessing the risk of comorbidities should be noted. Using the raw patient counts, a higher frequency of cardio- or cerebrovascular disease was observed within the non-TRD group. However, using the adjusted model, it was found that the risk of cardio- or cerebrovascular disease was higher in the TRD group. This phenomenon can be explained by the fact that these disorders are commonly associated with old age, and that older patients were over-represented in the non-TRD group in the patient sample analysed.

We found that self-harming behaviour not resulting in death was significantly more frequent in the TRD group than in the non-TRD group (p<0.0001). These results support previous study findings demonstrating that self-harming not resulting in death is associated with AD treatment resistance [8,35,37,45,64,65]. At present, it is not clear whether a history of self-harming behaviour not resulting in death increases the risk of TRD when compared with no history of this behaviour [58].

To our knowledge, only two previous studies, one from Sweden and one from the US, have compared the all-cause mortality in patients with TRD versus non-TRD MDD [13,28]. Both studies reported 29–35% higher overall mortality in patients with TRD compared with non-TRD patients [13,28]. Our study demonstrated a 38% increase in mortality in the TRD group compared with the non-TRD group in an age- and gender-adjusted model (HR: 1.38; 95% CI: 1.21–1.57), which is similar to the increases in mortality reported in the two aforementioned studies. Reutfors et al. investigated external causes (including suicides and accidents; ICD-10 codes V01–Y98) and non-external causes (all other relevant ICD-10 codes) of death separately, and found that the risk of external causes of death was increased in patients with TRD, but there was no increase in the risk of non-external causes of death [13]. We were unable to compare external and non-external data in this study as the NHIF database does not capture cause of death in sufficient detail. It should be noted that TRD patients had a significantly higher risk for having comorbidities and self-harming behaviour all of which can contribute to the higher risk of mortality. Ours is an estimate of the HR that is not independent of the effect of comorbidities and self-harming behaviour.

Similar to studies conducted in other countries [49,51,52], our study from Hungary observed that SSRIs were the most frequently prescribed drugs for the whole cohort. SSRIs were prescribed for the overwhelming majority of the non-TRD group, whereas non-SSRI ADs (e.g. mirtazapine and venlafaxine) were frequently prescribed for the TRD group.

As TRD was defined as the failure of at least two different ADs and/or add-on agents in this study, the higher number of different types of ADs and add-on medications (i.e. atypical antipsychotics, antiepileptics, thyroid hormones, lithium and buspirone) used by the TRD group compared with the non-TRD group during the study is in line with expectations. Furthermore, this supports the results reported by other studies [8,49,51,58,66].

## Study limitations

Our study has several limitations; primarily, the patient selection criteria were not designed to include an MDD population that is as broad as possible, but were designed to define a cohort in which pharmacotherapy-resistant depression (TRD) could be ascertained with high confidence. Firstly, the list of excluded medications (S3 Table in S2 File) may have removed some patients with MDD from the patient pool. For example, a patient may have been prescribed an anxiolytic—such as clonazepam in our case—by their GP prior to their referral to a psychiatrist and the subsequent prescription of an AD. However, patients who were prescribed an AD and concomitant anxiolytics, were included in the study. Additionally, the excluded diagnoses

(S2 Table in S2 File) may have also removed some patients with MDD, as some diagnoses, such as alcohol use disorders (F10), are known to be associated with MDD [67].

According to current knowledge, AD treatment of depressive episodes in the context of certain somatic disorders (e.g. Alzheimer's disease or malignancies) is not effective [68,69]. As we have excluded at least a proportion of the patients who had depression due to another medical condition ('organic depression') it is possible that the proportion of TRD detected in this study was underestimated.

With regard to the comorbidity-related analyses, in most cases, it was not possible to determine whether somatic comorbidities associated with TRD began before or after the onset of depressive symptoms. Therefore, it is impossible to determine whether the presence of TRD increases the risk of somatic comorbidities, or whether the reverse is true. The only exceptions are thyroid disorders that resulted in hypothyroidism, as patients who received thyroid hormone supplementation prior to the study period were excluded.

The way in which MDD index episodes were identified is a limitation of this study, whereby in some patients with prior AD exposure (before 2007, for which no data were available) the depressive episode was designated to be a new case of MDD. A patient is classified as 'new' at the time of their first AD prescription, which means that there is prior AD-free observation period. However, this observation period is not the same length for all patients. This period is the shortest for patients with an MDD index date at the beginning of 2009, for whom only a 2-year observation period was available (2007 to 2008), whereas a 7-year observation period was available for those with an MDD index date in 2014. The longer this observation period is, the lower the chance that a patient with prior AD exposure could be identified as new. Therefore, as time increases, the proportion of such patients in the newly identified patient pool decreases, which also caused the overall number to decrease. Based on the data available, it cannot be determined if the number of truly new patients increased, decreased, or remained stable during the study period.

The categorisation of patients as TRD or non-TRD was based on the number of prescribed ADs. This assumes that each AD treatment was discontinued due to a lack of efficacy. As this categorisation method did not take into consideration that lack of efficacy could have resulted from partial adherence or non-adherence, there could have been a degree of overestimation of the proportion of patients with TRD.

Only data on MDD subjects who were treated with antidepressant pharmacotherapy were available, while MDD subjects treated exclusively with other treatment modalities (e.g. psychotherapy, electroconvulsive treatment) were not available. For this reason: 1) the MDD population cannot be considered as complete; 2) the frequency of other kinds of TRD (other than pharmacotherapy resistant) could not be assessed.

Another bias of our study comes from the fact that shorter time-periods between the entry of patients into the study and the end of the observation period are expected to be associated with proportionally less chances of antidepressant change(s) and to become qualified as a treatment resistant case.

As cause of death data were unavailable, the exact nature of the excess mortality observed in the TRD group could not be established—this warrants further investigation.

## Conclusion

The proportion of patients with TRD within our MDD cohort (~8.3%) was consistent with similar previous studies in other countries. The majority of our other main findings (e.g. more frequent self-harming behaviour, increased risk of comorbid "Neurotic, stress-related and somatoform disorders" and higher overall mortality in TRD subjects) are also in line with

previous results from other countries. Taking the substantial proportion of patients with TRD into consideration, a more comprehensive and targeted treatment strategy would be required for these individuals.

## Supporting information

**S1 File. Minimal dataset derived from the database.** Sheet "Baseline, age"–Gender distribution, follow-up time, age distribution of patients; derivation of average age. Sheet "New patients by year"–Number of newly identified patients in the MDD and TRD patient groups. Sheet "Mortality"–Survival curve data, OS, TRD and non-TRD patients; Cox regression results. Sheet "Comorbidities1"–Baseline characteristics of patients involved in comorbidities analysis; raw frequencies of comorbidities. Sheet "Comorbidities2"–Results of logit regression on comorbidities. Sheet "Treatment patterns"–Distribution of patients with different number of treatment ingredients for the whole study period; frequency of usage of drug classes; top 10 most commonly used antidepressants.
(XLSX)

**S2 File.**
(DOCX)

**S3 File.**
(DOCX)

## Acknowledgments

The authors thank István Bitter, Department of Psychiatry and Psychotherapy, Semmelweis University, Budapest, Hungary for the helpful comments during data interpretation and manuscript preparation.

Editorial assistance in the development of this manuscript was provided by Open Health Medical Communications (UK).

## Author Contributions

**Conceptualization:** Péter Takács, László Fehér, Zoltán Rihmer.

**Data curation:** Péter Kunovszki, Péter Takács, Tamás Balázs, Károly Dede.

**Formal analysis:** Péter Kunovszki, Tamás Balázs, Károly Dede.

**Funding acquisition:** Péter Takács.

**Investigation:** Péter Döme, László Fehér, Siobhán Mulhern-Haughey, Sébastien Barbreau, Zoltán Rihmer.

**Methodology:** Péter Kunovszki, Péter Takács, Tamás Balázs, Károly Dede.

**Project administration:** Péter Kunovszki.

**Software:** Tamás Balázs, Károly Dede.

**Supervision:** Péter Takács.

**Validation:** Zoltán Rihmer.

**Visualization:** Péter Kunovszki.

**Writing – original draft:** Péter Döme, Péter Kunovszki, László Fehér, Tamás Balázs, Károly Dede, Zoltán Rihmer.

**Writing – review & editing:** Péter Döme, Péter Kunovszki, Péter Takács, László Fehér, Tamás Balázs, Károly Dede, Siobhán Mulhern-Haughey, Sébastien Barbreau, Zoltán Rihmer.

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
