## [Decision Letter · Decision Letter 0]

7 Nov 2020

PONE-D-20-32053

Clinical characteristics of treatment-resistant depression in adults in Hungary: real-world evidence from a 7-year-long retrospective data analysis

PLOS ONE

Dear Dr. Kunovszki,

Thank you for submitting your manuscript to PLOS ONE. After careful consideration, we feel that it has merit but does not fully meet PLOS ONE’s publication criteria as it currently stands. Therefore, we invite you to submit a revised version of the manuscript that addresses the points raised during the review process.

Two reviewers addressed several major and minor concerns about your manuscript. Please revise your manuscript carefully.

We look forward to receiving your revised manuscript.

Kind regards,

Kenji Hashimoto, PhD

Academic Editor

PLOS ONE

Journal Requirements:

2.Thank you for providing the following Funding Statement: 

[The study was funded by Janssen: Pharmaceutical Companies of Johnson & Johnson (www.janssen.com).

PK, LF, SMH, SB are employees of Janssen, PT was an employee of Janssen during the time the study was carried out.

Editorial assistance in the development of this manuscript was provided by Open Health Medical Communications (UK), with financial support from Janssen. The authors retained full editorial control over the content of the manuscript and the decision to publish it.

The funders had no role in study design, data collection and analysis, decision to publish, or preparation of the manuscript.].

We note that one or more of the authors is affiliated with the funding organization, indicating the funder may have had some role in the design, data collection, analysis or preparation of your manuscript for publication; in other words, the funder played an indirect role through the participation of the co-authors.

If the funding organization did not play a role in the study design, data collection and analysis, decision to publish, or preparation of the manuscript and only provided financial support in the form of authors' salaries and/or research materials, please review your statements relating to the author contributions, and ensure you have specifically and accurately indicated the role(s) that these authors had in your study in the Author Contributions section of the online submission form. Please make any necessary amendments directly within this section of the online submission form.  Please also update your Funding Statement to include the following statement: “The funder provided support in the form of salaries for authors [insert relevant initials], but did not have any additional role in the study design, data collection and analysis, decision to publish, or preparation of the manuscript. The specific roles of these authors are articulated in the ‘author contributions’ section.”

If the funding organization did have an additional role, please state and explain that role within your Funding Statement.

Please also provide an updated Competing Interests Statement declaring this commercial affiliation along with any other relevant declarations relating to employment, consultancy, patents, products in development, or marketed products, etc.  

3.We note that you have indicated that data from this study are available upon request. PLOS only allows data to be available upon request if there are legal or ethical restrictions on sharing data publicly. For information on unacceptable data access restrictions, please see http://journals.plos.org/plosone/s/data-availability#loc-unacceptable-data-access-restrictions.

Reviewers' comments:

Reviewer's Responses to Questions

**Comments to the Author**

1. Is the manuscript technically sound, and do the data support the conclusions?

Reviewer #1: Yes

Reviewer #2: Partly

2. Has the statistical analysis been performed appropriately and rigorously? 

Reviewer #1: Yes

Reviewer #2: Yes

3. Have the authors made all data underlying the findings in their manuscript fully available?

Reviewer #1: No

Reviewer #2: Yes

4. Is the manuscript presented in an intelligible fashion and written in standard English?

Reviewer #1: Yes

Reviewer #2: Yes

5. Review Comments to the Author

Reviewer #1: This is a very well-written manuscript. I read this study with great interest. Although this study used claimed data, which have some inherent disadvantage, the authors have done their best to optimize the robustness of their findings. The TRD decision algorithm is reasonable and convincing. The authors also reported a balanced consideration in terms of the limitations. Data presented in tables and figures are informative. I have only some minor concerns for the authors' reference.

1. In the introduction section, it would be better to prepare the readers the rationale to examine the relationship between TRD and mortality risk.

2. In the methodology section, the reader was not informed for how to link mortality data for each participant. In many countries, the mortality data is independent from other databases. As per line 430, it seems that mortality data are also part of NHIF database in Hungary. Please clarify this point early in the methodology section.

3. Is it reasonable to exclude individuals who use buspirone (Table S3)? Buspirone should be an anxiolytics not antidepressant. Similarly, why some BZD, such as clonazepam, lorazepam, are categorized as class I antidepressant in Hungary (Table S5)?

4. In table S4, why conversion(dissociative disorder) and somatoform disorders are categorized as comorbid anxiety disorders? These disorders should be different from anxiety spectrum disorders, both in DSM and ICD system.

5. In lines 229-231, the authors were suggested to make it clear regarding why to use match method to avoid bias in the analysis of mortality risk. What kind of bias? Besides, in lines 231, the TRD and non-TRD individuals were matched as per age and sex. However, in line 310, the legend for figure 4 denoted that they were matched by "date". Please clarify this.

6. Table 2 is not easy to interpret. Why did not the authors use traditional odds ratio and its 95% CI to present their data? Log (OR) is circular.

7. In the analysis of mortality risk, why did the authors only specify age and sex as covariate in the Cox model? Obviously, as per the finding of this study, somatic comorbiditis are more prevalent in TRD patients. If the effect of somatic disorders is not controlled for, the relationship between TRD and elevated mortality risk is only a confounding effect of comorbid somatic disorders. If the authors did not intend to emphasize the independent effect of TRD on morality risk, they should explain that in the manuscript.

Reviewer #2: This observational retrospective epidemiological study aimed to examined the proportion of treatment-resistant depression (TRD) in a cohort of patients with major depression in Hungary (n=99531). By utilizing a nationwide longitudinal database, they found that the proportion of TRD patients in Hungary was about 8.3%. The prevalence of psychiatric and somatic comorbidities and self-harming behaviors also increased in TRD patients.

In general, it is an interesting and well-executed study. However, a few methodological points may need to be further clarified to avoid adding confusion before its publication.

1. To answer the above questions, I don't know if it is necessary to recruit patients with a new onset of depressive episode. But since you have mentioned about this, please provide reference for you to use the operational criteria to make sure it is a validated method.

2. It looks like many patients with depression were excluded. How many patients (%) with a diagnosis of major depression were excluded ? Is that possible the exclusions make the results un-applicable to clinical settings. Please specify and discuss potential bias/limitation from that.

3. Patients with major depression were included if they had at least 2 AD prescriptions within 6 months at any time during the study. The criteria may fail to recruit a proportion of depression patients who were not depressed enough to initiate a AD drug. If so, the identified TRD ratio may be overestimated in this regard. In addition, if the subject with major depression was identified close to the study end point, there might be no enough time period for us to identify whether the subject is TRD or not, using the criteria you used to define TRD.

4. Certain somatic diagnosis and medications led to the exclusion (line 174-195). You also addressed it in the limitation. However, one the major purposes of this study was to examine the somatic comorbidity of the MDD patients ? Potential limitation on that should also be addressed.

5. It mentions in line 235: "A prespecified list of comorbidities was analysed in a fixed 2-year time period, which started on the TRD index date for patients with TRD and on the MDD index date for patients without TRD." Is this a validated method? Such definitions may lead to unfair comparisons, since the exact time to be analysed for the occurrence of comorbidities is different. I will suggest to investigate all the comorbidities after the index date until the end of the follow-up (e.g., Li et al., Major Depressive Disorder and Stroke Risks: A 9-Year Follow-Up Population-Based, Matched Cohort Study. PLoS One. October 8, 2012 ).

5. A recent study reported that TRD also increased the risks of dementia and you may consider to add the citation (Yee-Lam E Chan et al., Treatment-Resistant depression enhances risks of dementia and Alzheimer's disease: A nationwide longitudinal study. J Affect Disord. 2020 Sep 1;274:806-812.)

6.I agree that there is not yet a standardised definition for TRD and existing definitions have key differences, but do not totally agree with the paragraph about TRD in the introduction. In fact, there is no consistent definition regarding TRD or treatment-refractory depression. Some even used medication-resistant depression to describe these patients. Therefore, it is incorrect to say in line 52 that TRD is a term used to describe a subpopulation of MDD with a suboptimal response to "an adequate dose and duration" of antidepressants (ADs) and treatment refractory depression refers to an episode of MDD with insufficient response to numerous sequential AD trials (line 62). It is suggested to tone down a bit and to emphasize a wide range of TRD with poor responses to treatment in clinical settings and no consistent definitions across studies about TRD.

7. The study needs to be reexamined by a native English speaker. Some typos could be seen throughout the paper. (e.g., line 60: a least  at least)

6. PLOS authors have the option to publish the peer review history of their article (what does this mean?). If published, this will include your full peer review and any attached files.

Reviewer #1: No

Reviewer #2: No

---

## [Author Response · Author response to Decision Letter 0]

7 Dec 2020

We have updated the style of the manuscript, including reference formatting.

2. Funding statement and Competing interests

The funding organization only contributed to the study in the form of providing salaries to Janssen employees, providing funds for the data access fee for data, providing funds for Heatlthware Ltd. for participation, and providing funds for the medical writer.

In accordance with this we would like to update the funding statement to the following:

“The study was funded by Janssen: Pharmaceutical Companies of Johnson & Johnson (www.janssen.com).

PK, LF, SMH, SB are employees of Janssen, PT was an employee of Janssen during the time the study was carried out.

Janssen provided funds for Healthware Ltd. for their participation in the study. TB, KD are employees of Healthware Ltd. 

Editorial assistance in the development of this manuscript was provided by Open Health Medical Communications (UK), with financial support from Janssen. The authors retained full editorial control over the content of the manuscript and the decision to publish it.

The funder provided support in the form of salaries for authors PK, LF, SMH, SB, PT, provided funds for the data access, provided funds to Healthware Ltd., and provided funds for medical writing. The funder did not have any additional role in the study design, data collection and analysis, decision to publish, or preparation of the manuscript. The specific roles of these authors are articulated in the ‘author contributions’ section.”

I have checked the Author Contributions in the submission system, and they are correct.

We would like to update the Competing Interests Statement to the following:

“I have read the journal’s policy and the authors of this manuscript have the following competing interests:

PK, LF, SMH and SB are employees of Janssen.

PT was formerly an employee of Janssen.

TB, KD are employees of Healthware Ltd., which company received funding from Janssen for the participation in the study.

PD received fees for consultancy from Janssen.

ZR received fees for consultancy from Janssen, received speaker’s honoraria from Janssen, Servier and Krka, and served as an advisory board member for Lundbeck, Janssen, Servier and Krka.

This does not alter our adherence to PLOS ONE policies on sharing data and materials.”

3. Data access

The dataset that was used is held by the National Health Insurance Fund (NHIF) of Hungary (http://www.neak.gov.hu, e-mail: neak@neak.gov.hu).

Access to the individual-level data is available after filing a formal data access request to adatkeres@neak.gov.hu. Requestors need to accept the terms and conditions of the data request and may need to pay the corresponding data access fee.

The terms of the contract for data access does not allow the reporting of any data of a single individual or results which comes from aggregating the data of less than 10 individuals. Therefore, a de-identified dataset could not be provided. Taking these requirements into consideration, the results can be published.

A supplementary dataset was created which contains the patient counts derived from the original data.

Reviewers’ comments:

Reviewer #1:

1. In the introduction section, it would be better to prepare the readers the rationale to examine the relationship between TRD and mortality risk.

Although, we have mentioned in the Introduction section (lines 69-70 in the first version of the MS.), that according to the results of the few extant studies there is a link between treatment resistance and increased mortality in MDD, as per this request of Reviewer#1, we have completed this part of the text in the revised version with a short discussion on the association between depression (as a whole) and increased mortality/decreased life expectancy. In fact, with the analysis of the effect of TRD on mortality we would like to contribute to the scarce existing findings on this association.

2. In the methodology section, the reader was not informed for how to link mortality data for each participant. In many countries, the mortality data is independent from other databases. As per line 430, it seems that mortality data are also part of NHIF database in Hungary. Please clarify this point early in the methodology section.

In the Data source section it was mentioned that the database “includes demographic data, such as date of birth, geographical region, gender, and date of death”, but we agree that the fact that the database includes death information needs to be highlighted more. For this reason, this section was rephrased, and the death information availability was also added to the Study measures section as well.

3. Is it reasonable to exclude individuals who use buspirone (Table S3)? Buspirone should be an anxiolytics not antidepressant. Similarly, why some BZD, such as clonazepam, lorazepam, are categorized as class I antidepressant in Hungary (Table S5)?

We would like to thank Reviewer#1 for this valuable comment.

In fact, in our study we considered buspirone – similarly to lithium, thyroid hormones, antiepileptics and second generation antipsychotics – as an agent for augmentation of an antidepressant* (in other words as an “add-on” agent – see lines 226-227 of the first MS version) and not as an anxiolytic agent. Since TRD was defined as the failure of at least two different ADs and/or “add-on” agents in the study (see lines 442-445 of the first MS version) we thought it is logical to exclude not only subjects with antidepressant prescription between 01 January 2007 and 31 December 2008 (a two-year long period before the study period [i.e. 01 January 2009 and 31 August 2015]) but also those who received add-on agents (including buspirone) anytime during a 180-days period before their MDD index date (first MS version, lines 180-182). We have also made a footnote in S3 Table that medications are only used as exclusion criteria during this time period for clarity purposes.

In the supplementary document it was incorrectly identified that the financial guideline referenced therein was only considering antidepressants. In fact, the guideline covers all medications which are used in the treatment of “Mood, neurotic, stress-related, somatoform disorders, and bulimia nervosa”. We have modified the supplementary document accordingly. We have also removed drugs from the table (S8 Table) which were not identified as either ADs or add-on medications in the study.

It is important to highlight that all PTDs of patients had to start with an AD, so for example it is not possible that a patient is on clonazepam monotherapy and the failure of clonazepam is assessed as such.

*Bauer M et al. Pharmacological treatment of unipolar depressive disorders: summary of WFSBP guidelines. Int J Psychiatry Clin Pract. 2017;21:166-176.

4. In table S4, why conversion (dissociative disorder) and somatoform disorders are categorized as comorbid anxiety disorders? These disorders should be different from anxiety spectrum disorders, both in DSM and ICD system.

We very much appreciate this critic of Reviewer#1. In fact, we used the F4X main group of the ICD-10 that is referred to “Neurotic, stress-related and somatoform disorders”. In addition to several classic anxiety disorders (various phobic disorders, panic disorder, generalized anxiety disorder, mixed anxiety and depressive disorder etc.), this group consists actually of some disorders not belonging to anxiety disorders (e.g. somatoform and dissociative disorders) and also some ailments (OCD and PTSD) that were considered as anxiety disorders in the DSM-IV (but are not yet in the DSM-5). Due to this critic of Reviewer#1, we have replaced the term “anxiety”/“anxiety disorders” to “Neurotic, stress-related and somatoform disorders” along the MS. On the other hand, we would like to mention that, for obvious reasons, Hungarian National Health Insurance Fund uses the ICD (and not the DSM) for coding various somatic and mental disorders thus we were constrained to use ICD codes.

5. In lines 229-231, the authors were suggested to make it clear regarding why to use match method to avoid bias in the analysis of mortality risk. What kind of bias? Besides, in lines 231, the TRD and non-TRD individuals were matched as per age and sex. However, in line 310, the legend for figure 4 denoted that they were matched by "date". Please clarify this.

We thank Reviewer#1 for pinpointing that these parts of the methods were not described in enough detail. Using this matching technique, TRD patients are matched to other MDD patients who are not TRD at the same time (measured from their own MDD index date). If TRD patients had only been matched to non-TRD patients a selection bias would have arisen due to excluding patients based on future events (the patients who are not yet TRD but become one at a later time). We have added these pieces of information to the revised MS.

The other point also highlights a sentence that could be easily misunderstood. Matched date refers to the date when the (at that time) non-TRD patients are matched to the TRD patients. This is now clarified in the Methods section.

6. Table 2 is not easy to interpret. Why did not the authors use traditional odds ratio and its 95% CI to present their data? Log (OR) is circular.

According to the suggestion of Reviewer#1 we have changed the table which now shows the OR and 95%CI of OR.

7. In the analysis of mortality risk, why did the authors only specify age and sex as covariate in the Cox model? Obviously, as per the finding of this study, somatic comorbidities are more prevalent in TRD patients. If the effect of somatic disorders is not controlled for, the relationship between TRD and elevated mortality risk is only a confounding effect of comorbid somatic disorders. If the authors did not intend to emphasize the independent effect of TRD on morality risk, they should explain that in the manuscript.

We thank Reviewer#1 for this comment. As per the suggestion of the Reviewer we did not intend to obtain an independent effect of TRD on the mortality. This was clarified in the Discussion section of the revised MS. While obtaining an independent estimate would be interesting, the complexity of such analysis is out of scope for the current study.

Reviewer #2:

1. To answer the above questions, I don't know if it is necessary to recruit patients with a new onset of depressive episode. But since you have mentioned about this, please provide reference for you to use the operational criteria to make sure it is a validated method.

We would like to thank Reviewer#2 for this comment. Actually, in our opinion, it would be impossible to establish a pharmacotherapy resistant depressive episode (as we, similarly to others, defined it in this paper) if subjects with ongoing depressive episodes at the beginning of the observation period had been recruited, since, in that case, we would be unable to say how many antidepressants (or add-on agents) were previously (i.e. before the start of the observation period) administered to the given patient for the treatment of his/her current (i.e. ongoing) episode. As a consequence, for instance, it would be undecidable whether a given subject whose depression had begun before the observation period already has fulfilled the criteria of TRD when he/she entered into the study.

2. It looks like many patients with depression were excluded. How many patients (%) with a diagnosis of major depression were excluded? Is that possible the exclusions make the results un-applicable to clinical settings. Please specify and discuss potential bias/limitation from that.

We think that in the Study limitations subchapter we firmly emphasized that due to the application of various exclusion criteria probably not all TRD patients were included (for example, TRD cases due to some kind of somatic comorbidity were excluded). Exclusion criteria served the purpose of identification and exclusion of “pseudo”-TRD patients (e.g. those who have a somatic comorbidity or alcohol abuse that may contribute to the “resistance”). Thus, exclusions were not autotelic and with their use we were able to catch more precisely those cases where TRD was not the consequence of a depression provoking/aggravating condition.

Due to technical limitations only a crude estimate of 40% can be given for the proportion of patients excluded. The four most common reasons for exclusion were: diagnosis ICD F06 (Other mental disorders due to brain damage and dysfunction and to physical disease), diagnosis ICD F31 (Bipolar affective disorder), diagnosis ICD F10 (Mental and behavioural disorders due to use of alcohol), prescription of levothyroxine sodium in the baseline period. Considering the first three out of these four criteria, we may suppose with great certainty that those who were excluded due to these actually did not have MDD, but a major depressive episode in the context of another psychiatric disorder (i.e. "organic" brain disorder; bipolar disorder; substance-induced depression) than MDD. To sum up, although at first glance the exclusion of 40% of the possible MDD patients seems high their exclusion was absolutely justified.

3. Patients with major depression were included if they had at least 2 AD prescriptions within 6 months at any time during the study. The criteria may fail to recruit a proportion of depression patients who were not depressed enough to initiate a AD drug. If so, the identified TRD ratio may be overestimated in this regard. In addition, if the subject with major depression was identified close to the study end point , there might be no enough time period for us to identify whether the subject is TRD or not, using the criteria you used to define TRD.

Reviewer#2 is right that those MDD subjects who received other treatment modalities than pharmacotherapy (psychological therapy, electroconvulsive treatment, repetitive transcranial magnetic stimulation) were not included in the study. Accordingly, we completed the Study limitations section of the revised MS with this fact.

Reviewer#2 is also right when states that shorter time-periods between the entry of patients into the study and the end of the observation period are associated with proportionally less chances of antidepressant change(s) and, in this manner, fulfilling the requirements of TRD for the given patients. In a novel part of the Study limitations subchapter we discuss this issue.

4. Certain somatic diagnosis and medications led to the exclusion (line 174-195). You also addressed it in the limitation. However, one the major purposes of this study was to examine the somatic comorbidity of the MDD patients? Potential limitation on that should also be addressed.

We would like to thank this comment of Reviewer#2. However, we also would like to draw his/her attention that exclusion diagnoses (see them listed in Appendix Document 1) do not overlap with the analyzed comorbidities (see these listed in Appendix Document 3).

5. It mentions in line 235: "A prespecified list of comorbidities was analysed in a fixed 2-year time period, which started on the TRD index date for patients with TRD and on the MDD index date for patients without TRD." Is this a validated method? Such definitions may lead to unfair comparisons, since the exact time to be analysed for the occurrence of comorbidities is different. I will suggest to investigate all the comorbidities after the index date until the end of the follow-up (e.g., Li et al., Major Depressive Disorder and Stroke Risks: A 9-Year Follow-Up Population-Based, Matched Cohort Study. PLoS One. October 8, 2012 ).

We thank Reviewer#2 for this comment. We agree that involving more comorbidities would enrich the results of the study, however the complexity of adding new comorbidities now is out of scope for the current study.

In the time-independent model using different lengths of follow-up for patients would cause biases in the analysis. Considering two patients with the same probability of having a certain comorbidity, the patient with the longer follow-up time would have a higher probability of being identified as having the comorbidity. We used uniform length follow-up periods to combat this bias and excluded any patients who would be lost to follow-up before reaching this length of observation.

6. A recent study reported that TRD also increased the risks of dementia and you may consider to add the citation (Yee-Lam E Chan et al., Treatment-Resistant depression enhances risks of dementia and Alzheimer's disease: A nationwide longitudinal study. J Affect Disord. 2020 Sep 1;274:806-812.)

In accordance with this request of Reviewer#2, we inserted the paper suggested as a new reference into the text (see lines 418-419 in the clean version of the revised MS).

7.I agree that there is not yet a standardised definition for TRD and existing definitions have key differences, but do not totally agree with the paragraph about TRD in the introduction. In fact, there is no consistent definition regarding TRD or treatment-refractory depression. Some even used medication-resistant depression to describe these patients. Therefore, it is incorrect to say in line 52 that TRD is a term used to describe a subpopulation of MDD with a suboptimal response to "an adequate dose and duration" of antidepressants (ADs) and treatment refractory depression refers to an episode of MDD with insufficient response to numerous sequential AD trials (line 62). It is suggested to tone down a bit and to emphasize a wide range of TRD with poor responses to treatment in clinical settings and no consistent definitions across studies about TRD.

Reviewer#2 is right that we should emphasize more strongly that in the current study we investigated pharmacotherapy-resistant depression in stricto sensu (and not TRD in genere). Accordingly, we rephrase the first sentence of the second paragraph of the “Introductionˮ section. Furthermore, in line with the suggestion of Reviewer#2, the incriminated sentence on "treatment refractory" depression has been deleted from the revised version.

8. The study needs to be reexamined by a native English speaker. Some typos could be seen throughout the paper. (e.g., line 60: a least  at least)

As per this suggestion of Reviewer#2, we thoroughly proofread the text and corrected the typos in it.

---

## [Decision Letter · Decision Letter 1]

2 Jan 2021

Clinical characteristics of treatment-resistant depression in adults in Hungary: real-world evidence from a 7-year-long retrospective data analysis

PONE-D-20-32053R1

Dear Dr. Kunovszki,

We’re pleased to inform you that your manuscript has been judged scientifically suitable for publication and will be formally accepted for publication once it meets all outstanding technical requirements.

Kind regards,

Kenji Hashimoto, PhD

Section Editor

PLOS ONE

Additional Editor Comments (optional):

Reviewers' comments:

Reviewer's Responses to Questions

**Comments to the Author**

1. If the authors have adequately addressed your comments raised in a previous round of review and you feel that this manuscript is now acceptable for publication, you may indicate that here to bypass the “Comments to the Author” section, enter your conflict of interest statement in the “Confidential to Editor” section, and submit your "Accept" recommendation.

Reviewer #1: All comments have been addressed

Reviewer #2: All comments have been addressed

2. Is the manuscript technically sound, and do the data support the conclusions?

Reviewer #1: Yes

Reviewer #2: Yes

3. Has the statistical analysis been performed appropriately and rigorously? 

Reviewer #1: Yes

Reviewer #2: Yes

4. Have the authors made all data underlying the findings in their manuscript fully available?

Reviewer #1: Yes

Reviewer #2: Yes

5. Is the manuscript presented in an intelligible fashion and written in standard English?

Reviewer #1: Yes

Reviewer #2: Yes

6. Review Comments to the Author

Reviewer #1: I read this manuscript again and ensure that the authors have adequately addressed my comments . I think this manuscript is now suitable to be published in PLOS ONE.

Reviewer #2: (No Response)

7. PLOS authors have the option to publish the peer review history of their article (what does this mean?). If published, this will include your full peer review and any attached files.

Reviewer #1: No

Reviewer #2: **Yes: **Cheng-Ta Li

---

## [Editor Report · Acceptance letter]

8 Jan 2021

PONE-D-20-32053R1 

Clinical characteristics of treatment-resistant depression in adults in Hungary: real-world evidence from a 7-year-long retrospective data analysis 

Dear Dr. Kunovszki:

I'm pleased to inform you that your manuscript has been deemed suitable for publication in PLOS ONE. Congratulations! Your manuscript is now with our production department. 

Kind regards, 

on behalf of

Prof. Kenji Hashimoto 

Section Editor

PLOS ONE